# Iron whiskers on asteroid Itokawa indicate sulfide destruction by space weathering

Toru Matsumoto [1,2 ✉], Dennis Harries[2], Falko Langenhorst[2,3], Akira Miyake[4] & Takaaki Noguchi[1]

Extraterrestrial iron sulfide is a major mineral reservoir of the cosmochemically and astro-biologically important elements iron and sulfur. Sulfur depletion on asteroids is a long-standing, yet unresolved phenomenon that is of fundamental importance for asteroid evolution and sulfur delivery to the Earth. Understanding the chemistry of such environments requires insight into the behavior of iron sulfides exposed to space. Here we show that troilite (FeS) grains recovered from the regolith of asteroid 25143 Itokawa have lost sulfur during long-term space exposure. We report the wide-spread occurrence of metallic iron whiskers as a decomposition product formed through irradiation of the sulfide by energetic ions of the solar wind. Whisker growth by ion irradiation is a novel and unexpected aspect of space weathering. It implies that sulfur loss occurs rapidly and, furthermore, that ion irradiation plays an important role in the redistribution of sulfur between solids and gas of the interstellar medium.

[1] Faculty of Arts and Science, Kyushu University, 744 Motooka, Nisi-ku, Fukuoka 819-0395, Japan. [2] Institute of Geosciences, Friedrich Schiller University Jena, Carl-Zeiss-Promenade 10, 07745 Jena, Germany. [3] Hawai'i Institute of Geophysics and Planetology, School of Ocean and Earth Science and Technology, University of Hawai'i at Manoa, Honolulu, HI 96822, USA. [4] Division of Earth and Planetary Sciences, Kyoto University, Kitashirakawaoiwake-cho, Sakyo-ku, Kyoto-shi 606-8502, Japan. ✉email: matsumoto.toru.502@m.kyushu-u.ac.jp

Energetic ions from various cosmic sources can significantly alter solid mineral grains in interplanetary and interstellar space[1–3]. Within the solar system, solar wind ions are considered to be a main causes of space weathering, which alters the physical and chemical surface properties of airless bodies such as the Moon and asteroids[1,2]. Iron sulfides such as troilite (FeS) are common minerals in solar system materials. Although iron sulfides are important solid reservoirs of the cosmically abundant elements sulfur and iron, their behavior in space-exposed environments is poorly constrained. Remote-sensing of the S-type asteroid 433 Eros indicates significant sulfur depletion in its surface regolith by space weathering of iron sulfides[4–7]. One of the fundamental chemical characteristics of airless bodies are their volatile element abundances, including sulfur, which closely correlate with elemental fractionation during solid accretion in the solar nebula, subsequent thermal processing and gravitational differentiation[8,9]. Moreover, the sulfur abundance of small airless bodies is of great importance to estimate the amount of sulfur delivered to the early Earth, which likely became a major ingredient of a habitable surface and life[10]. However, the sulfur depletion by space weathering may obscure the detection of original sulfur abundances of airless bodies. Thus, the rates and modes of space weathering of iron sulfides are essential to understand the dynamics of asteroids, their relics of the early solar system and their impact on the terrestrial planets and life. Understanding the alteration of minerals by the solar wind is important for unveiling their surface evolution, because the altered regolith grains are key indicators of the timescales of geological processes on these dynamical bodies, such as motion, loss and replenishment of the regolith[11,12]. Furthermore, knowledge of the interaction between solids and energetic ions may help to understand the chemistry of cosmic radiation environments in the interstellar medium (ISM)[3,13], where molecular clouds and new stars are born and deliver chemical components to emerging planetary systems, such as the early solar system.

Regolith particles recovered from the S-type asteroid 25143 Itokawa by the Hayabusa mission are key samples to examine the space weathering of primordial minerals, especially the effects of solar wind implantation[2,12,14–18]. Their mineralogy corresponds to LL4-6 ordinary chondrites[19,20], in which troilite is the major sulfide phase. The few previous studies of iron sulfides have reported little modifications by the solar wind[15,16] compared to the distinctly radiation-damaged rims of silicate minerals[2,14], whereas earlier experimental studies have suggested chemical and structural changes of iron sulfides by ion irradiation[21–23]. So far, naturally altered iron sulfide has not been comprehensively examined. Here we present systematical analyses of the surface features of troilite-bearing Itokawa particles. Field emission scanning electron microscopy (SEM), focused ion beam (FIB) sectioning and transmission electron microscopy (TEM) are used to better understand the surface modifications of space-exposed troilite (see the Methods section, Supplementary Notes 1–4, and Supplementary Figs. 1–3). We show that the wide-spread occurrence of metallic iron whiskers as a decomposition product formed through solar wind irradiation on the troilite. It implies that sulfur loss occurs rapidly on airless bodies and, furthermore, that ion irradiation plays an important role in the partitioning of sulfur and iron in interplanetary and interstellar space, if small dust particles are involved.

## Results

### The characteristics of troilite-bearing Itokawa particles. A striking feature of the troilite surfaces is the appearance of elongated whiskers of metallic iron protruding from the space-

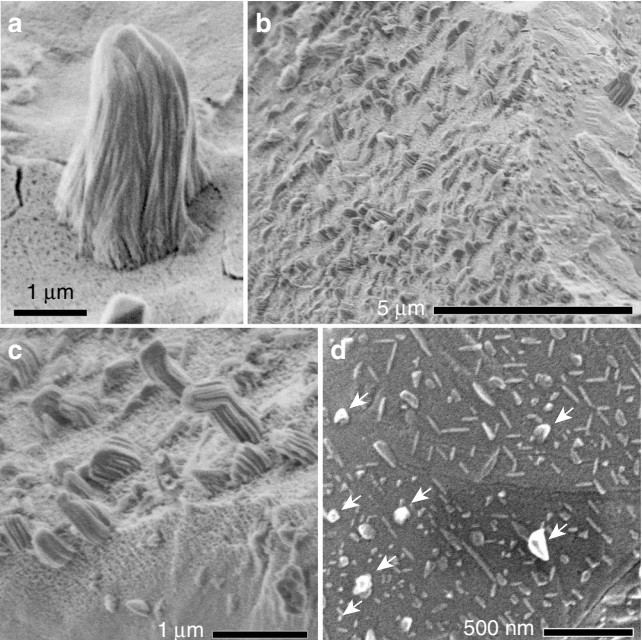

**Fig. 1 Surface features of space-exposed troilite in Itokawa samples.**
**a-c** Secondary electron (SE) images of whiskers on iron sulphide surfaces at different scales. Dark dots on the iron sulphide surface in (**a**) and (**c**) are open vesicles. A gradual change was observed from a vesicular texture to a non-vesicular texture in (**c**). **d** SE image of thin plates elongated in specific directions, which apparently correlate with crystallographic directions of the iron sulphide substrate. Whiskers are indicated by arrows. **a-c** shows surfaces of particle RA-QD02-0325. **d** is a surface of particle RA-QD02-0302 (see the sample list in Supplementary Table 1).

exposed surfaces (Fig. 1). They are present on 8 of 10 Itokawa particles we have investigated, suggesting their common appearance (Supplementary Table 1, Supplementary Note 1). The whiskers are thinner at their tips, curved, and striated. Their length varies from several tens of nm to 2.8 µm. The whiskers often coexist with thin flat plates that are deficient in sulfur and possibly composed of iron (Fig. 1d). Troilite surfaces that have whiskers often exhibit vesicular textures (Fig. 1c, Supplementary Table 1).

TEM observation showed that each whisker appears to be composed of a set of several sub-whiskers (Fig. 2a). Selected area diffraction (SAED) patterns of the whisker confirmed that they consist of bcc-structured metallic iron (Fig. 2b). Individual sub-whiskers show the same diffraction contrast in TEM images (Fig. 2a), suggesting that single crystals of iron are predominant within them. SAED patterns and diffraction contrast of thinner whiskers (<300 nm in width) even show that they are composed of almost one single iron crystal (Fig. 2e) or several crystals with slightly different crystallographic orientations. The metal of the whiskers is typically almost pure iron (<0.7 wt% Ni), only in a few cases minor Ni was concentrated near the uppermost surfaces of some sub-whiskers (~4.1 wt%; Fig. 2d, Supplementary Table 2).

Open vesicles on troilite surfaces are delimited by subparallel, scale-like structures (Fig. 3a), which are elongate perpendicular to the c-axis of the troilite (Fig. 3a, b). Dark-field TEM images of the vesicular regions show many bright spots corresponding to small crystalline sub-domains (>10 nm) having slightly different crystallographic orientations compared to the substrate troilite (Fig. 2f). This occurs within 80–100 nm in depth. In the case of troilite apparently showing a non-vesicular texture in SEM images, many internal vesicles appear beneath the troilite surface, which extend to ~50 nm in depth (Fig. 3c). Relatively large

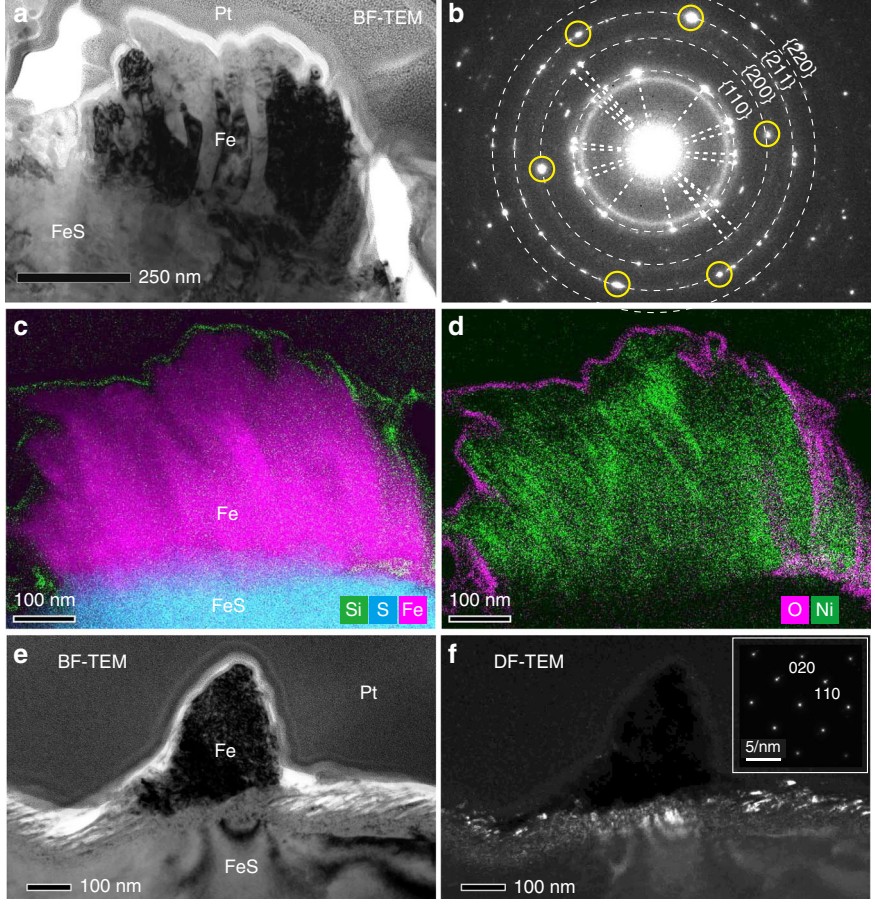

**Fig. 2 Detailed structures of metallic iron whiskers on troilite. a** A TEM bright-field (BF) image of a whisker on troilite in a FIB section. **b** A SAED pattern of the whole region of the whisker. Dotted rings have radii corresponding to the d-spacings of {hkl}planes of bcc iron. Pairs of diffraction spots from the same crystals are connected by dotted lines. The six circled spots are reflected from a single iron crystal belonging to a zone axis [$\bar{1}$20]. **c-d** X-ray intensity maps of the whisker in (**a**). Composite image of Si (green), S (cyan), and Fe (magenta) is shown in (**c**) and that of O (magenta) and Ni (green) is displayed in (**d**). **e** A TEM-BF image of a thinner whisker in a FIB section from an edge-on view (the electron beam was nearly parallel to the troilite surface). **f** A TEM-dark field (DF) image corresponding to (**e**). A SAED pattern obtained from the whisker is shown on the upper right, showing bcc iron in zone axis [001]. **a-d** was obtained from FIB section RA-QD02-0292-01 (Supplementary Fig. 2). **e-f** was obtained from FIB section RA-QD02-0325-01 (Supplementary Fig. 3).

vesicles are concentrated at depths between 10 and 20 nm (Fig. 4b). SAED patterns of troilite in the vicinity of the vesicles indicate slight crystallographic misorientations (Fig. 3d). No discernable relationship in orientation between the iron whiskers and the troilite is evident in SAED patterns. The Fe/S ratio increases in the surface region with open vesicles as compared with the bulk troilite (Fig. 4a). In contrast, the Fe/S ratio of the surfaces with internal vesicles appears to be constant (Fig. 4b).

The whiskers, troilite, and silicate surfaces are covered with O- and Si-rich layers that are 7–15 nm thick (Fig. 2c, d). Such layers on troilite contain O, Si, and Mg (Fig. 4), and in some cases, Na, Cl, and Ca. These rims are probably vapor-deposited coatings generated by solar wind sputtering and/or micrometeorite impacts on nearby regolith grains[2]. The accumulation of vapor-deposited materials appears to have occurred after whisker growth. The olivine and low-Ca pyroxene coexisting with troilite have outermost vapor-deposited rims and lower polynanocrystalline rims that are generally interpreted as products of solar wind damage[2,14] (Supplementary Fig. 4). The residence times of the examined Itokawa grains within a few millimeters of the uppermost regolith surface were estimated to be ~1000 to ~8000 yr, calculated from solar flare track densities in silicates[24] (Supplementary Table 1 and Supplementary Fig. 5).

## Discussion

Polynanocrystalline rims and vesicles are known to be typical damage features of experimentally ion-irradiated materials[25] and solar-wind-irradiated minerals[2,14]. Vesicles might expand through the coalescence of implanted atoms and/or by the clustering of induced vacancies[26]. Expansion of vesicles along [uv0] crystallographic directions and/or simultaneous surface sputtering by ion implantation likely produced the open vesicles on Itokawa troilite. Troilite possesses a strong anisotropy between the [001] and [uv0] directions due to its hexagonal NiAs-type base structure, which quite certainly enhances elongation of open vesicles. The solar wind is a stream of charged ions from the Sun, composed mainly of $^1$H (~95.4%) and $^4$He (~4.6%) with typical kinetic energies peaking at ~1 keV/nucleon[27]. The depth of vesicles in the troilite (up to 40–50 nm) broadly matches the penetration depth of 4 keV He$^+$ of 47 nm[28], whereas large vesicles (Fig. 4b) in troilite are located at the penetration depth of 1 keV H$^+$ of 22 nm (Supplementary Fig. 6). These depths support the notion that vesicles developed through accumulation of solar wind gases.

Here we discuss the feasibility of iron whisker growth from the liquid, vapor and solid states. Aqueously mediated formation of iron whiskers appears highly unlikely, because of the dry

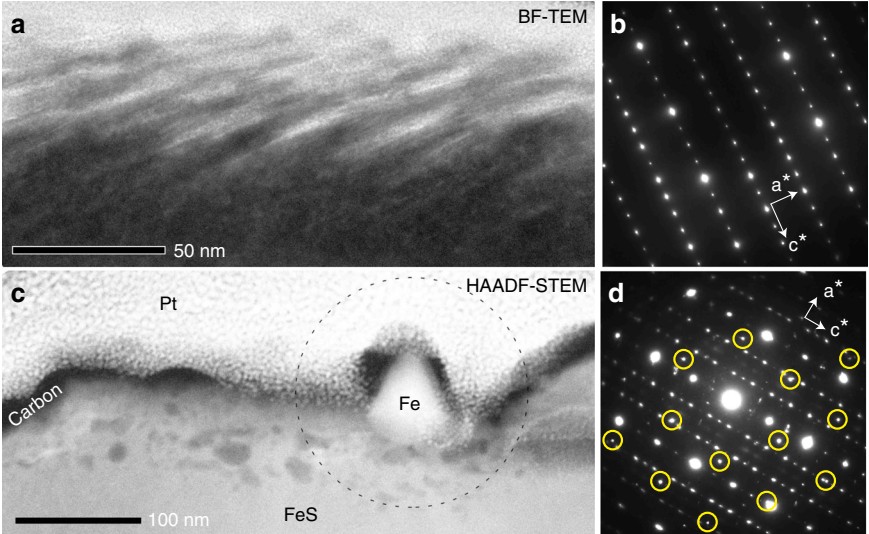

**Fig. 3 Structures of altered rims on troilite. a** A TEM-BF image of a troilite surface with open vesicles when viewed edge-on. **b** A SAED pattern of troilite obtained from the same sample direction as (**a**). **c** A HAADF-STEM image of troilite with internal vesicles. Dark patches beneath the troilite surface correspond to vesicles. An iron whisker is present on the troilite. **d** A SAED pattern from the circled area in (**c**). Spots from a single bcc iron having a zone axis of [111] are circled. The troilite diffraction pattern is slightly diffuse due to misorientation. **a-b** was obtained from a FIB section RA-QD02-0325-01 (Supplementary Fig. 1). **c-d** was obtained from FIB section RA-QD02-0286-01 (Supplementary Fig. 4).

conditions present on Itokawa and the instability of metallic iron under hydrous conditions. Vapor condensation may have occurred within fractures or pores in the host rocks of the parent body of Itokawa during thermal metamorphism[29]. Magnetite whiskers found in Martian meteorites were likely precipitated with carbonates from a supercritical fluid in fractures[30,31]. As for Itokawa grains, the walls of fractures/pores that formed during metamorphic equilibration can be identified as euhedral crystals on grain surfaces[12]. Although such equilibrated textures exist on the troilite surface (Supplementary Table 1), they do not correlate with whisker distribution. The majority of Itokawa grains display fractured surfaces that formed by the fragmentation of larger regolith grains[12], observed troilite surfaces are no exception (Fig. 1). Hence, whiskers on the troilite must have developed after the formation of the regolith grains. If iron whiskers were produced by condensation from a vapor, then Itokawa's young formation age of ~1.3–2.3 Ga[32–34] excludes the solar nebular gas (~4.5 Ga ago) as a vapor source being in contact with Itokawa's surface regolith. The origin of iron whiskers is therefore not comparable to that of graphite whiskers reported in CV chondrites, which are thought to have condensed from the nebular gas[35]. Impact-induced vaporization on Itokawa[2,14,15,17] is the most likely source of a vapor. However, assuming that the deposited rims have the bulk composition of LL chondrites, in which iron sulfides and metals account for only 6% of the total volume[36], the Si-bearing rims on Itokawa grains do not balance the widespread occurrence of iron whiskers on troilite, because these rims are quite thin (<15 nm). Additionally, the whiskers do not occur on silicates and do not show any structures suggesting preferential iron condensation on troilite, such as droplets necessary for vapor–liquid–solid (VLS) growth[37] or epitaxy[31]. Moreover, heating of LL-chondrite materials forms sulfur-rich vapors that cause sulphurization rather than condensation of metallic iron[38]. Therefore, it appears unlikely that the whiskers are condensed materials. The space-exposed surface on Itokawa may contrast with lunar surface environments, where abundant melts and vapor-deposited materials (up to 200 nm thickness) cover the regolith grains[39,40]. Graphite whiskers[41], optically transparent whiskers[42], and elongated iron objects called stalks[43]

have been reported in lunar samples, all of which were regarded as condensation products. Although the iron stalks consist of metallic iron, they do not correspond to iron whiskers on Itokawa, because they clearly show a droplet at their tip indicating VLS growth.

The remaining and most plausible mechanism for iron whisker formation is growth from a solid substrate that is supersaturated with cations, as proposed for the metallic whisker growth in sulfides or oxides[44–46]. This inference is supported by the fact that the iron whisker growth appears to be associated with the progressive irradiation-damage, given the lack of whiskers on troilite with minor damage[16]. Metallic iron may form when troilite comes into contact with $H_2$ gas[47]. The $H_2$ gas trapped in the vesicles may cause the reaction, $FeS + H_2 = H_2S + Fe^{2+} + 2e^-$ under solar heating (Supplementary Note 5, Supplementary Fig. 7). Shifting the equilibrium towards $H_2S$ requires removal of the $H_2S$ produced from the vesicle. This process takes place most efficiently where an escape path for $H_2S$ gas is available such as along sub-grain boundaries, cracks, or open vesicles, while $H_2$ is replenished by solar wind irradiation. Additional heating by micrometeorite bombardment and larger impact events may play a role for increasing the replacement of these gases by accelerating the diffusion rate of hydrogen atoms and $H_2S$ in troilite. Furthermore, these heating events may raise the reaction rate and subsequent diffusion of iron atoms. The acceleration of the reaction can also happen by solar heating when the asteroid has an orbit with a lower heliocentric distance[16]. The action of the solar wind can selectively sputter sulfur atoms. The S/Fe ratio of troilite decreased to $0.55 \pm 0.55$ after experimental 4 keV He irradiation[7] with a dose of $17 \times 10^{17}$ ions/cm$^2$, which corresponds to an exposure time of $4 \times 10^4$ yr to the solar wind at 1 AU. Energy transfer to sulfur atoms could result in the sulfur escape in the form of H-, S-bearing radicals and/or molecules[48,49].

Briefly, we discuss the possibility that the solar wind has not been involved in the metallic whisker growth from troilite: Possible processes to form excess iron are heating events by solar irradiation, micrometeorite bombardments, and larger impacts. The oxidation of sulfide in a H-free environment by solar heating results in incongruent evaporation of sulfur[50] ($FeS = 0.5S_2 +$

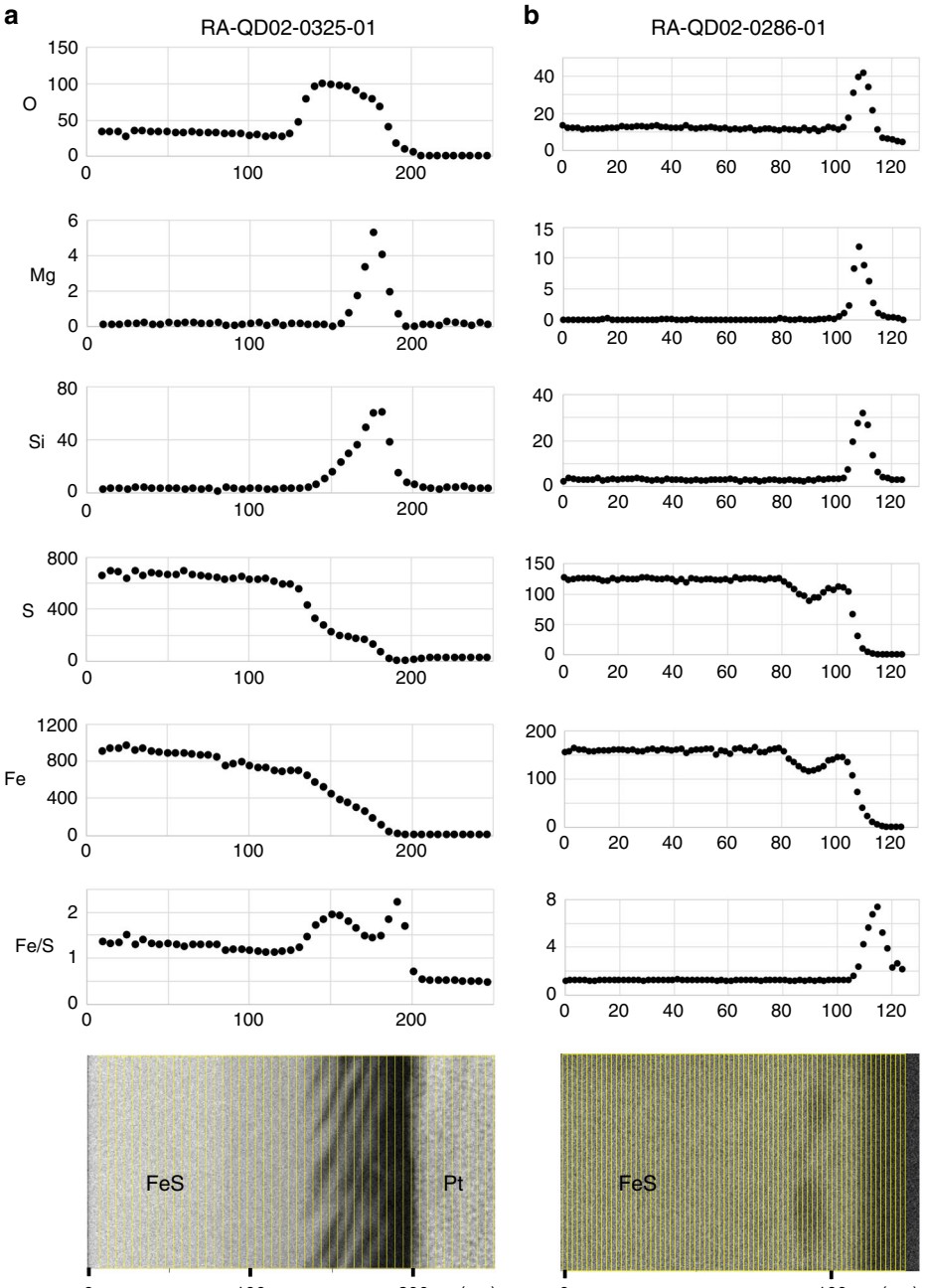

**Fig. 4 Depth profiles of major elements detected near the troilite surface.** X-ray intensity counts of O, Mg, Si, S, Fe and the count ratio of Fe/S near the troilite surface with (**a**) open vesicles and with (**b**) internal vesicles are shown. The vertical axis is X-ray count and the horizontal axis is distance (nm). The lower subfigure is a BF-STEM image of investigated area corresponding to upper count profiles. **a** was obtained from a FIB section RA-QD02-0325-01 (Supplementary Fig. 1). **b** was obtained from FIB section RA-QD02-0286-01 (Supplementary Fig. 4).

$Fe^{2+} + 2e^-$). Assuming that $S_2$ escape from the space-exposed troilite surface occurs at zero ambient $S_2$ partial pressure, the Hertz–Knudsen–Langmuir relationship results in an estimated a maximum evaporative flux of ~$3 \times 10^{-3}$ S atoms/cm²/yr at the maximum surface temperature of Itokawa of 130 °C (Supplementary Note 6). This corresponds to ~$1 \times 10^{-18}$ nm/yr, which is an extremely low rate. Submicron-sized impact craters on troilite in Itokawa particles show no evidence for whisker growth[51]. Moreover, examined troilite surfaces with iron whiskers do not correspond to crater floors. Somewhat larger impacts might locally induce a transient temperature rise of the regolith and could increase the equilibrium pressure of $S_2$, such that thermal evaporation could become more efficient. However, this process should take place at relatively greater depths in the regolith, where the temperature increases and the permeability required for $S_2$ loss are low. In addition, the temperature increase would be rather short in duration and long-term temperatures at depth are much lower than at the surface. Increased solar heating at possible lower heliocentric distances[16] might be insufficient for a fast recession of the troilite surface (Supplementary Note 6). Hence, selective sulfur loss appears not to be caused solely by heating events on the asteroid. Based on the discussion above, sulfur loss and supersaturation of $Fe^{2+}$ just beneath the surfaces of troilite grains very likely occurred as a consequence of solar wind

irradiation on Itokawa. Heating events might have contributed to promote the chemical reaction and/or atomic and molecular diffusion in troilite. The iron-supersaturated zone may have grown within the irradiation-damaged rim by rapid diffusion of excess iron atoms along defects. The excess $Fe^{2+}$ accumulated at a suitable metallic nucleus, where $Fe^{2+}$ was finally reduced to the metallic state ($Fe^0$) by electrons moving freely, given the good electrical conductivity of troilite ($Fe^{2+} + 2e^- = Fe^0$). Continued sulfur loss and the reduction of $Fe^{2+}$ caused the growth of the metallic nucleus. Surface diffusion of iron atoms might have contributed to form the cone shape of whiskers. The increase of surface energy of metals by whisker growth might be compensated for by relaxation of the strain field associated with excess interstitial $Fe^{2+}$. Another important factor for whisker growth might have been the relaxation of the compressive stress[52] in metal due to the difference in thermal expansion between troilite and metal induced by thermal cycling during the rotation of Itokawa.

From a calculation of the total whisker volume per unit area of troilite, the production rate of excess iron was estimated to be ~0.04 nm/yr or ~0.13 troilite monolayers/yr assuming a space-exposure age of $10^3$ yr (Supplementary Note 7; Supplementary Fig. 8; Supplementary Table 3). The average characteristic length of iron diffusion in troilite on Itokawa's surface in its present-day orbit is the order of 10 nm/yr on average (Supplementary Note 8; Supplementary Fig. 9). This indicates that whisker growth is not controlled by cation diffusion, but rather by the production of excess cations. The rapid iron diffusion enables accumulation of sufficient excess $Fe^{2+}$ for whiskers growth. Hence, the total volume of whiskers at a given surface area might be a novel proxy for the duration of regolith space exposure that corresponds to the amount of excess iron accumulated. The net growth mechanism of iron whiskers on Itokawa appears to be unique among the extraterrestrial whiskers, which have been widely considered as vapor condensates[30,31,35,41–43,53].

Solar wind irradiation has had a limited effect on the sulfur abundance on the surface of Itokawa, because the estimated sulfur loss is negligible compared with the total troilite volume. This is probably due to a short exposure time. Therefore, it is interesting that sulfur depletion was detected in the uppermost 10 μm of regolith on the S-type asteroid Eros that occupies heliocentric distances similar to Itokawa. Deeper sulfur loss is possible but not accessible to remote X-ray analysis[54]. Adopting the sulfur loss rate due to solar wind on Itokawa (0.04 nm/yr), sulfur depletion to 10 μm depth takes about $2.5 \times 10^5$ yr. Given that the regolith ages of Eros is 3–50 million years[55], this indicate that substantial amounts of sulfur could have been lost through regolith gardening[5]. Sulfur loss through other processes, such as impact vaporization[4–6] may have occurred additionally. The solar wind may be most effective on bodies at low heliocentric distances, where the increased ion flux and surface temperature can exponentially enhance $H_2$ formation and sulfur loss. Alteration rims on iron sulfide might provide a clue for past orbital states of airless bodies with secular variations of their perihelia, as proposed for silicates[16]. Effective solar wind irradiation may also occur in small interplanetary dust due to their high surface-to-volume ratio. Sulfur loss by the irradiation could contribute to the sulfur abundance of the Earth, if irradiated small grains were involved in accretion of the Earth[56].

Our observations have strong implications for the alteration of iron sulfides in the interstellar environments. Iron sulfides have been detected in the outflows of evolved stars[57] and are likely injected into the interstellar medium (ISM)[58]. An unsolved question regarding the chemical composition of the ISM is the unique role of sulfur in the gas phase. In the diffuse ISM sulfur is barely depleted from the gas phase[59,60]. On the contrary, in dense molecular clouds sulfur is strongly depleted in the gas, which has

been attributed to complex astrochemical reactions involving organo-sulfur compounds[61]. Here it is not entirely clear how sulfur can be transferred between inorganic sulfides, the gas and, eventually, organics in these environments. Submicron-sized silicate dust in the diffuse ISM has been considered to become amorphous by the irradiation of $H^+$ and $He^{++}$ accelerated by shockwaves from supernovae[3]. The flux of ions with energies of 1 keV/nucleon in the shockwave models ($10^7$–$10^9$ ions/cm$^2$/s) and their total ion dose (up to $10^{18}$–$10^{19}$ $H^+$ ions/cm$^2$)[13] are similar to the solar wind (flux[62] = $2 \times 10^8$ $H^+$ ions/cm$^2$/AU$^2$/s, dose = $6 \times 10^{18}$ $H^+$ ions/cm$^2$, assuming an exposure age of $10^3$ yr at 1 AU). Our observation supports the hypothesis[60] that sulfur is supplied to the gas phase of the ISM via the destruction of iron sulfides due to ion irradiation. Rapid whisker growth as opposed to the formation of a protective iron shell implies that sulfides are easily destroyed under such conditions. Metallic iron is predicted to appear as a byproduct of this alteration process in the ISM and might exists there even though its primary formation by homogeneous nucleation from supernova ejecta appears to be highly ineffective[63]. Metallic iron in the ISM can be an important catalysts for molecule formation[64] and can produce magnetic dipole emissions that disturb the cosmic microwave background[65]. Therefore, the alteration of iron sulfides by ion implantation could have a large impact on the chemistry and electromagnetic properties of the ISM.

From our study, we conclude that, contrary to previous evidence, iron sulfides are highly susceptible to decomposition by ion irradiation in space. Sulfur loss from iron sulfides by solar wind could cause a remarkable decrease of sulfur abundances on airless bodies at low heliocentric distances. Their susceptibility to this process implies that ion irradiation plays an important role for the partitioning of iron and sulfur in interplanetary and interstellar environments. Iron whiskers are newly recognized features of space weathering and could become distinctive indicators to interpret the history of airless bodies, such as the C-group asteroids currently studied by Hayabusa2 and OSIRIS-REx. The appearance of whiskers induced by ion irradiation has never before been found in the fields of mineralogy and material science —potentially their further study will provide new insights into whisker growth mechanisms and corrosion processes due to ion irradiation.

## Methods

**Samples**. The Itokawa particles investigated in this study were collected from room A of the sample catcher and were captured during the second touchdown in the MUSES-C Region on Itokawa. The mineral phases and average diameters of the Itokawa particles were investigated during their initial description[66] at the Extraterrestrial Sample Curation Center (ESCuC) of the Japan Aerospace Exploration Agency (JAXA).

**SEM analysis**. We examined surface features of 10 regolith particles (average diameters = 68–241 μm) including iron sulfide (Supplementary Table. 1) using a field emission SEM (FE-SEM; Hitachi SU6600) at the ESCuC after the initial routine description. The particles were placed on an Au-coated holder for SEM observations, without a conductive coating, using an electrostatically controlled micromanipulation system. Until these SEM observation, the Itokawa particles had never been exposed to an atmospheric environment, thus minimizing the contamination and alteration of the studied particles[66]. We performed secondary electron (SE) imaging at accelerating voltages of 1.5 and/or 2.0 kV under high vacuum with an electron beam current of ~10 pA.

Three Itokawa particles (RA-QD02-0286, RA-QD02-0292, and RA-QD02-0325) were transferred onto an adhesive carbon-conductive tape for further analysis. We determined the elemental compositions of their surfaces with an energy-dispersive X-ray spectrometer (EDX) using an FE-SEM (Hitachi SU6600) equipped with a X-Max$^N$ 150 mm$^2$ (Oxford Instruments) in JAXA and an FE-SEM (Hitachi SU6600) equipped with a Bruker XFlash® FlatQUAD detector at the Institute for Molecular Science (IMS, Higashi-Okazaki, Japan). The accelerating voltage for SE imaging was 1.5 kV, whereas for EDS analysis we used 5 and 10 kV.

**TEM analysis**. We then extracted electron-transparent sections for transmission electron microscopy (TEM) studies of regions of interest on iron sulfide surfaces, using focused ion beam (FIB) systems. The FIB systems were a FEI Quanta 200 3DS at Kyoto University, Japan, and a FEI Quanta3D FEG at the University of Jena, Germany. To protect the particle surfaces during FIB processing and enhance the electrical conductivity, the particles were coated with carbon (~20 nm thick). We then coated the particle surfaces with an electron-beam-deposited Pt layer (at 5 kV) followed by a Ga ion-beam-deposited Pt layer (at 30 kV). For each particle, sections that were a few tens of micrometers in size were extracted using a microsampling manipulator and mounted onto TEM copper grids. After the target samples were attached to the TEM grids, they were thinned to ~100 nm using a 30 kV $Ga^+$ beam and were finally cleaned using a 5 kV $Ga^+$ beam at 40–80 pA.

Thin sections from the three particles were labeled RA-QD02-0286-01, RA-QD02-0292-01, and RA-QD02-0325-01. These sections were prepared in order to sample surface features of the iron sulfide initially detected by SEM, and also typically included also space-exposed silicates. We extracted an additional section (RA-QD02-0325-02) from pyroxene next to the iron sulfide sampled by RA-QD02-0325-01, because the latter did not sample any exposed silicates.

Prepared sections were examined using a FE-TEM (FEI Tecnai $G^2$ FEG) equipped with an Oxford 80 mm$^2$ energy-dispersive SDD X-ray detector and a Gatan UltraScan 2k CCD camera at the Institute for Geosciences, University of Jena, Germany. EDX analysis was performed in scanning TEM (STEM) mode aided by high-angle annular dark-field (HAADF) imaging. Quantitative elemental abundances within the whisker metal were calculated using the Cliff–Lorimer thin film approximation with $k$-factors determined from an artificial Fe–Ni–Co–Cr alloy of known composition[67]. High-resolution elemental profiles (Fig. 4) were obtained by acquiring laterally extended multispectral X-ray images in STEM mode, followed by segmenting, lateral integration, and fitting of extracted X-ray spectra. These spectra were evaluated semi-quantitatively to provide X-ray intensities and intensity ratios across surface layers with spatial resolutions of a few nanometer in the direction perpendicular to the surface. Compared with spot measurements, this approach minimizes beam damage. We performed further quantitative EDX analysis of the iron sulfides using a FE-TEM (JEOL JEM-3200FSK) equipped with an EDX detector at Kyushu University, Kyushu, Japan. The elemental compositions of the iron sulfides were calculated using the Cliff–Lorimer thin film approximation. We used troilite nodules in the Cape York iron meteorite and a terrestrial millerite from Sanany, Ural in Russia, as standards for the $k$-factors. EDX detection limits depend on the counting statistics and we achieved down to 0.1 wt% for Cr, Ni and P in metal and sulfide (Cr and P were not detected at this level). Co and Cu have substantially higher detections limits (at least one order of magnitude) due to overlap of X-ray lines and spurious contributions from the sample environment (TEM column and Cu TEM grids).

Densities of solar flare tracks (SFT) in silicate grains of each particle were determined using low-angle annular dark-field (LAADF) imaging in the STEM mode at a long camera length. SFTs were counted in the resulting images and we estimated the ratio of SFTs to the visible area of the track-sensitive mineral.

## Data availability
The data that support this study are available from the corresponding author upon request.

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

## Acknowledgements

We appreciate the JAXA curation team for giving us an opportunity to study Itokawa samples and for assisting the study. We also thank Dr. Satoshi Nakao and Dr. Masahiro Sakai for supporting SEM observations in the IMS. The present study is supported by Grant-in-Aid for JSPS Fellows (18J00579) and JSPS Early Career Scientists (18K13610), and the JSPS core to core program: International Network of Planetary Sciences. F.L. thanks the Deutsche Forschungsgemeinschaft for funding the SEM/FIB and TEM facilities at the University of Jena via the Gottfried Wilhelm Leibniz prize (LA830/14-1).

## Author contributions

T.M. designed the research, carried out the analysis, and wrote the manuscript. D.H. processed the FIB sectioning and performed TEM observations at the University of Jena. F.L. provides advice on the interpretation of the TEM analysis. A.M. assisted FIB sample preparation at Kyoto University. T.N. supported TEM analysis at Kyushu University. All authors contributed to write and edit the manuscript.

## Competing interests

The authors declare no competing interests.
