## [Peer Review File · Nature Communications]

Reviewers' comments:

Reviewer #1 (Remarks to the Author):

The authors present interesting new observations of and interpretations for iron whiskers on regolith particles returned from the asteroid Itokawa. No previous observations of iron whiskers on planetary materials have been reported. The finding of iron whiskers and their related impact on deciphering environmental conditions on asteroid surfaces and big-picture connection to solid-gas interactions in the interstellar medium are novel and will be of interest to a broad community.

The major claims of the paper are that troilite grains in regolith particles from the asteroid Itokawa have lost sulfur due to energetic solar wind ion irradiation (e.g., space weathering). The loss of sulfur has caused decomposition of FeS to Fe, where Fe grows to form whiskers. Moreover, this process might play an important role in the distribution of Fe and S in the interstellar medium (ISM).

The compelling data and science interpretation presented in the paper warrant publication in *Nature Communications*.

There are a few edits and comments that I recommend the authors and editor consider making before publication, all of which are minor.

1) It is my recommendation is that the paper should focus more on the discovery of these iron whiskers on planetary materials. The discovery of these unique features (and the value of planetary sample return) gets lost because the paper puts more attention on the inference of what these iron whiskers can tell us about the ISM. I would also ask that the authors take greater lengths to show that the whiskers are not some kind of terrestrial alteration product. My request stems from the idea that many Itokawa grains have been previously studied and none have shown these whiskers, and now this study finds 8 out of 10 grains show these whiskers? While there might not be much room in the main manuscript, perhaps the supporting information could discuss why whiskers haven't been seen before, what is special about these samples, sampling site location on the asteroid, and/or compositional evidence that these whiskers aren't some kind of post sampling alteration product.

2) Introduction: The paper reports the finding of iron whiskers on Itokawa regolith particles and then infers that the process that makes these Fe-whiskers could apply to processes in the ISM. Hence, the Introduction should start with a background on asteroid space weathering and information regarding Hayabusa sampling of Itokawa because it serves as the core of the paper. Once the reader is given the details surrounding the focus of the paper, only then should the authors discuss the bigger implications inferred from conclusions based on investigating the Fe-whiskers.

3) line 60, the paper uses two references by Christoffersen & Keller (2011) and Keller et al. (2010) to conclude that iron sulfides are highly radiation resistant minerals. However, it is shown in irradiation experiments that the relative sputtering rate of FeS is much higher than olivine or enstatite in a more recent abstract by Keller et al. 2013 (Keller, L., Rahman, Z., Hiroi, T., Sasaki, S., Noble, S., Horz, F., Cintala, M., 2013. Asteroidal space weathering: The major role of FeS (abstract# 2404), 44th Lunar and Planetary Science Conference. CD-ROM.). These experiments also showed that irradiation of FeS with 5kV Ga⁺ resulted in preferential sputtering of S and the formation of a 5-8-nm thick surface layer of nanophase Fe metal. Hence, the authors need to reconsider the statement that iron sulfides are highly radiation resistant. The authors also state that (lines 184-186) sulfides are easily destroyed under such conditions, conditions in the ISM that are similar to the solar wind. The authors need to reconcile this apparent discrepancy about whether FeS is radiation resistant or relatively easy alter.

4) The idea that "metallic iron may form when the troilite comes into contact with H₂ gas may not be the complete scenario. As discussed in Guo et al. 2019 (Guo, L., Zhong, S., Bao, Q., Gao, J., & Guo, Z. (2019). Nucleation and Growth of Iron Whiskers during Gaseous Reduction of Hematite Iron Ore Fines. *Metals*, 9(7), 750.) heating and reduction must occur for the formation of iron whiskers to occur. The authors of this paper appear too quickly to discount the need for heating to catalyze the reaction. For instance a pulse heating event would allow a "rapid iron diffusion" event, however, I do not envision constant ion bombardment providing a means for rapid iron diffusion. In addition, Guo et al. explain that substrate composition influenced iron whisker formation. The presence of SiO₂ decreased the total porosity, which stopped the near-surface nucleation of iron particles and thereby prevented the formation of iron whiskers. Could the process be that iron sulfides are heated/vaporized, S is lost as a vapor, and Fe recondenses preferentially on FeS grains? This would mean that the iron whiskers are not from the FeS grain they are in contact with. Such a scenario would circumvent apparent problems with micrometeorite heating because the FeS vapor forms in one location but is deposited in a different location, which might only be mm or 100 mm away from the impact or sputtered source. This situation would be similar to the Hapke 2001 formation of nanophase iron in the lunar regolith. Hapke, B., 2001. Space weathering from Mercury to the asteroid belt. *J. Geophys. Res.* 106, 10039-10073.

5) The Sulfur may not leave as S₂ but as H₂S and H₂S instead. Gillis-Davis, J.J., Zhu, C., Góbi, S., Abplanalp, M.J., Frigge, R., Kaiser, R.I., 2019. Laboratory Space Weathering Induced Formation of Sulfides in the Murchison (CM2) Meteorite, Lunar & Planetary Science Conference, abstract 1213.

6) line 162: The authors say that the degree of whisker growth (which to me implies whisker size and/or population) might be a novel proxy for the duration of regolith space exposure.... If, as the authors say and I agree, that whisker growth occurs by rapid growth, then it would seem that it is not the size of whisker growth but more specifically the population or number of iron whiskers that could be used as a proxy for the duration of space exposure, which is loosely analogous to lunar nanophase iron and its associated ferromagnetic measurement (FMR), which yields an accumulated exposure age as quantified by I_s/FeO.

7) Could the 7-15 nm layer of silicate be used to estimate the timing of formation of the Fe-whiskers? i.g., silicate rims on mineral grains are used to estimate exposure to space. Lunar grains in more mature soils have rims that are > 100 nm while in less mature soils they are <60 nm or even incomplete.

Fig 2f in the caption is as a darkfield image but high Z elements in darkfield images are bright. Both the Fe and the Pt are dark.

Fig 3c the caption says TEM-BF image but the image label states correctly that it is a HAADF image. Also in this image, the label for the scale bar is either too light to see or is not there. And last, the caption for image 3d says the SAED pattern from the circled area in 3c but there is no area in 3c circled that I can see.

Suggestions for different word choices: iron monosulphide can just be called iron sulfide (abstract); relicted is a better word for heirloom (introduction line 50), which is used to describe grains leftover from the early solar system; (line 106) the authors use the word segregation but I think they mean coalescence; lines 113-114 the authors use the terms vesicles and cavities as if they refer to different physical properties. If the two terms mean the same thing (which I think they do) I would recommend the authors not say vesicular regions and cavities or vesicles and cavities but just say vesicles; I would recommend not using modifiers like very and heavily.

It would also be good to make Supplementary Fig. 1. much bigger. I could barely see the features.

Supplementary Fig. 4., I do not see where troilite is coexisting with olivine in the figure 4a, it

would be great if the image were annotated with rim and host olivine, also the context for the compositional should be given (i.e., where do those rectangular composition maps overlay on the sample). Because I can't tell where the x-ray data are from I don't know whether the C map shows a high C amorphous rim? or just the carbon coat?

Reviewer #2 (Remarks to the Author):

Review of Iron whiskers on asteroid Itokawa indicate sulphide destruction by space weathering – Matsumoto et al.

This manuscript presents important observations of the effects of space weathering on sulfide minerals returned by the JAXA Hayabusa spacecraft. These observations are important as they relate to the long-standing puzzle in planetary science about the depletion of sulfur observed on S-type asteroid surfaces, most notably the near-Earth asteroid (433) Eros characterized by the NASA NEAR-Shoemaker mission. The authors present a thorough and meticulous data set, including many outstanding images of the asteroid grain surfaces. The inferences about sulfur loss are important not only for understanding asteroid surfaces but also for understanding sulfide mineral evolution in the interstellar medium and protoplanetary disks. As such, this work has applicability to a broad range of space scientists and is suitable for publication in *Nature Communications*. It is well-written and organized. I have a few minor comments to help improve the manuscript.

Under the "Mechanisms of whisker formation" section, the authors quickly rule out formation by condensation. I suggest that this mechanism needs a more thorough treatment. The work cited in references 28-30 provides a solid foundation for their preferred mechanism. However, whisker growth via condensation has been discussed in the meteorite literature (e.g., Bradley, John P., Ralph P. Harvey, and Harry Y. McSween Jr. "Magnetite whiskers and platelets in the ALH84001 Martian meteorite: Evidence of vapor phase growth." *Geochimica et Cosmochimica Acta* 60.24 (1996): 5149-5155.; Fries, Marc, and Andrew Steele. "Graphite whiskers in CV3 meteorites." *Science* 320.5872 (2008): 91-93.; Bradley, John P., Donald E. Brownlee, and D. R. Veblen. "Pyroxene whiskers and platelets in interplanetary dust: evidence of vapour phase growth." *Nature* 301.5900 (1983): 473.). Furthermore, micrometeorite impacts into Itokawa grains are known to deposit energy into a small volume of the asteroid surface, causing melting, vaporization, ionization, and recondensation of the target and impactor material (M. S. Thompson et al., *Microchemical and structural evidence for space weathering in soils from asteroid Itokawa. Earth, Planets and Space* 66.1, 89 (2014)). Thus, it is not so straightforward to dismiss vapor growth as a potential source mechanism. The authors should at least reference this other work and compare it to the preferred whisker growth mechanism more thoroughly.

For the whisker composition, it would be useful to have the quantitative Ni data for all the whisker compositions. Also, I request that the authors include the Ni profiles in the whiskers that showed enrichment in this element. Other elements that are commonly associated with troilite in ordinary chondrites are Cr, P, Co, and Cu (e.g., Lauretta, D. S. "Sulfidation of an Iron-Nickel-Chromium-Cobalt-Phosphorus Alloy in 1% H₂-S-H₂ Gas Mixtures at 400-1000° C." *Oxidation of Metals* 64.1-2 (2005): 1-22.; Rubin, Alan E. "Metallic copper in ordinary chondrites." *Meteoritics* 29.1 (1994): 93-98.). For these other elements, it would be important to know if they were analyzed for, as well as their detection limits. These elements could provide important clues to the formation mechanism.

The authors discuss both solar-wind irradiation and solar heating as potential mechanisms for the production of the iron whiskers. These two mechanisms do not seem to be mutually exclusive. In particular, the sulfur evaporation rate is determined to assume a vacuum environment. However, this mechanism is likely to be enhanced by the presence of solar-wind implanted hydrogen. If it is not feasible to include the effect of hydrogen in the evaporation model, then this interaction should at least be discussed in the supplement section on this model.

Dante Lauretta

Reviewer #3 (Remarks to the Author):

Lines 23-24 (abstract) state "Whisker growth is a novel and unexpected aspect of space weathering." This is incorrect, whisker growth associated with space weathering has been widely reported, mostly in the context of space weathering on the lunar regolith, for the past 50+ years. Many papers are published on the topic yet none of them are cited in this manuscript. Perhaps the most relevant paper is by Carter 1973 who describes metallic iron whiskers on the moon, interpreted to result from micrometeoroid impacts.

Hibbs, A. R. A hypothesis that the surface of the moon is covered with needle crystals. *Icarus*, 2, 181-186 (1963).

Brownlee et al., Whiskers on the moon. In *Analysis of Surveyor 3 material and photographs returned by Apollo 12*. NASA SP-284, 295 pages, published by NASA, Washington, D.C., 1972, p.236

Carter, J. L. VLS (Vapor-Liquid-Solid): Newly Discovered Growth Mechanism on the Lunar Surface? *Science* 181, 841-842 (1973).

Steel, A. et al., Graphite in an Apollo 17 Impact Melt Breccia. *Science* 329, 51 (2010).

Comment. We wish to express our appreciation to the reviewers for the insightful comments. The comments have helped us to improve significantly the paper. Here is a point-by-point response to the reviewers. We have highlighted the changes within the manuscript.

Reviewer #1:

***Q.1.* It is my recommendation is that the paper should focus more on the discovery of these iron whiskers on planetary materials. The discovery of these unique features (and the value of planetary sample return) gets lost because the paper puts more attention on the inference of what these iron whiskers can tell us about the ISM.**

A.1. In accordance with the reviewer's comment, we have changed the introduction section. We removed detailed description about ISM and stressed the importance of space weathering in planetary science (lines 37-43). Implication of our study for interstellar environments were discussed in the last part of the discussion section (lines 209-228). In addition, we have changed the abstract to adapt the revised manuscript (line 19-20).

***Q.2.* I would also ask that the authors take greater lengths to show that the whiskers are not some kind of terrestrial alteration product. My request stems from the idea that many Itokawa grains have been previously studied and none have shown these whiskers, and now this study finds 8 out of 10 grains show these whiskers? While there might not be much room in the main manuscript, perhaps the supporting information could discuss why whiskers haven't been seen before, what is special about these samples, sampling site location on the asteroid, and/or compositional evidence that these whiskers aren't some kind of post sampling alteration product.**

A.2. In accordance with the reviewer's comments, we have added the argument against iron whiskers being alteration products during sample preservation in Supplemental discussion (lines 621-628). In addition, we have explained that very limited observation had been performed to iron sulphides so far. Our discovery was mainly resulted from the accumulation of Itokawa samples available for analysis due to continuous curatorial work in JAXA. Also, we have mentioned that our samples are not special among Itokawa samples (line 70, lines 630-645).

***Q.3. Introduction:* The paper reports the finding of iron whiskers on Itokawa**

regolith particles and then infers that the process that makes these Fe-whiskers could apply to processes in the ISM. Hence, the Introduction should start with a background on asteroid space weathering and information regarding Hayabusa sampling of Itokawa because it serves as the core of the paper. Once the reader is given the details surrounding the focus of the paper, only then should the authors discuss the bigger implications inferred from conclusions based on investigating the Fe-whiskers.

A.3. This comment may be the same as *Q.1*. We have changed the introduction section. We removed detailed description about ISM and stressed the importance of space weathering in planetary science. Implication for interstellar environments were discussed in the last part of the discussion section. In addition, we have changed our abstract to adapt the revised manuscript.

***Q.4.* line 60, the paper uses two references by Christoffersen & Keller (2011) and Keller et al. (2010) to conclude that iron sulfides are highly radiation resistant minerals. However, it is shown in irradiation experiments that the relative sputtering rate of FeS is much higher than olivine or enstatite in a more recent abstract by Keller et al. 2013 (Keller, L., Rahman, Z., Hiroi, T., Sasaki, S., Noble, S., Horz, F., Cintala, M., 2013. Asteroidal space weathering: The major role of FeS (abstract# 2404), 44th Lunar and Planetary Science Conference. CD-ROM.). These experiments also showed that irradiation of FeS with 5kV Ga⁺ resulted in preferential sputtering of S and the formation of a 5-8-nm thick surface layer of nanophase Fe metal. Hence, the authors need to reconsider the statement that iron sulfides are highly radiation resistant. The authors also state that (lines 184-186) sulfides are easily destroyed under such conditions, conditions in the ISM that are similar to the solar wind. The authors need to reconcile this apparent discrepancy about whether FeS is radiation resistant or relatively easy alter.**

A.4. Our statement could be confused. We have corrected our sentences as below;
Lines 56-58: "... , whereas, earlier experimental studies have suggested chemical and structural changes of iron sulphides by ion irradiation"

***Q.5.* The idea that "metallic iron may form when the troilite comes into contact with H₂ gas may not be the complete scenario. As discussed in Guo et al. 2019 (Guo, L., Zhong, S., Bao, Q., Gao, J., & Guo, Z. (2019). Nucleation and Growth of Iron**

Whiskers during Gaseous Reduction of Hematite Iron Ore Fines. Metals, 9(7), 750.) heating and reduction must occur for the formation of iron whiskers to occur. The authors of this paper appear too quickly to discount the need for heating to catalyze the reaction. For instance a pulse heating event would allow a "rapid iron diffusion" event, however, I do not envision constant ion bombardment providing a means for rapid iron diffusion.

A.5. We agree that heating events have to be considered in our reduction model. We have clarified the contribution of the heating events when solar wind hydrogen is accumulated in troilite using the Hertz-Knudsen-Langmuir model. Detailed calculation has been added in Supplemental discussion (Lines 647-710). As a result, we have added the following text;

Line 151-153: "Additional heating by micrometeorite bombardment and larger impact events may play a role for increasing the replacement of these gases by accelerating the diffusion rate of hydrogen atoms and H₂S in troilite. Furthermore, these heating events may raise the reaction rate and subsequent diffusion of iron atoms."

Line 175: "Heating events might have contributed to promote the chemical reaction and/or atomic and molecular diffusion in troilite"

On the other hand, we have mentioned that selective sulphur loss cannot be explained solely by heating events on the asteroid Itokawa (Lines 158-173).

Q.6. In addition, Guo et al. explain that substrate composition influenced iron whisker formation. The presence of SiO₂ decreased the total porosity, which stopped the near-surface nucleation of iron particles and thereby prevented the formation of iron whiskers.

A.6. Guo et al. and references therein described that the presence iron silicate during the reduction of SiO₂-doped wüstite initiated the iron whisker formation. This effect might have prevented whisker formation. On the other hand, the presence of CaO in wüstite promoted the whisker growth. Although these investigations would be important for a comprehensive understanding toward whisker growth, we cannot reveal these effects from our dataset in this study.

Q.7. Could the process be that iron sulfides are heated/vaporized, S is lost as a vapor, and Fe recondenses preferentially on FeS grains? This would mean that the iron whiskers are not from the FeS grain they are in contact with. Such a scenario would circumvent apparent problems with micrometeorite heating because the FeS vapor forms in one location but is deposited in a different location, which might only be mm or 100 mm away from the impact or sputtered source. This situation would be similar to the Hapke 2001 formation of nanophase iron in the lunar regolith. Hapke, B., 2001. Space weathering from Mercury to the asteroid belt. *J. Geophys. Res.* **106, 10039-10073.**

A.7. In accordance with the reviewer's comment, we have added the description discussing the possibility of whiskers being vapor deposited materials formed from various conditions (lines 117-141). Because of scarce vapour-deposition-layers on grains and the lack of structures suggesting preferential iron condensation, We have described that it appears unlikely that the whiskers are condensed materials.

Q.8. The Sulfur may not leave as S₂ but as H₂S and H₂S instead. Gillis-Davis, J.J., Zhu, C., Góbi, S., Abplanalp, M.J., Frigge, R., Kaiser, R.I., 2019. Laboratory Space Weathering Induced Formation of Sulfides in the Murchison (CM2) Meteorite, Lunar & Planetary Science Conference, abstract 1213.

A.8. We agree the importance of Gillis-Davis et al. (2019) and reference therein, which described molecule formation via recombination reactions of suprathemal atoms by ion implantation. Therefore, we have added the following text:

Line 158 "Energy transfer to sulphur atoms could result in the sulphur escape in the form of H-, S- bearing radicals and/or molecules"

However, it is unclear if their results can be directly adapted for space-exposed FeS of Itokawa grains. Gillis-Davis et al. (2019) performed irradiation experiments of carbonaceous chondrite with infrared pulsed laser and 5 keV electrons. The experiments simulated thermal effects of micrometeorite bombardments and secondary electrons produced during solar/cosmic ray implantation. PI-ReTOF-MS analysis resulted in the detection of H₂S and H₂S₂ released from irradiated samples. They inferred that these molecules likely formed by recombination reactions of suprathemal H and S atoms. We

expect that hydrogen atoms in the experiments may come from phyllosilicates and/or organic materials in the carbonaceous chondrite. This situation may be different from Itokawa's materials, which would have almost no water and organics.

Q.9. line 162: The authors say that the degree of whisker growth (which to me implies whisker size and/or population) might be a novel proxy for the duration of regolith space exposure.... If, as the authors say and I agree, that whisker growth occurs by rapid growth, then it would seem that it is not the size of whisker growth but more specifically the population or number of iron whiskers that could be used as a proxy for the duration of space exposure, which is loosely analogous to lunar nanophase iron and its associated ferromagnetic measurement (FMR), which yields an accumulated exposure age as quantified by I_s/FeO .

A.9. We agree that this point requires clarification. The I_s/FeO corresponds to the total volume of lunar nanophase iron normalized by FeO contents. Similarly, it is more accurate to say that the total volume of whiskers at a given surface area might be a proxy for exposure age, which includes the information of population, length, and thickness of the whiskers. Therefore, we have modified our description as below.

Line 192: "Hence, the total volume of whiskers at a given surface area might be a novel proxy for the duration of regolith space exposure ..."

Q.10. Could the 7-15 nm layer of silicate be used to estimate the timing of formation of the Fe-whiskers? i.g., silicate rims on mineral grains are used to estimate exposure to space. Lunar grains in more mature soils have rims that are > 100 nm while in less mature soils they are <60 nm or even incomplete.

A.10. Although vapor deposited materials (silicate amorphous + nanophase iron/FeS) might have accumulated with time, the deposition process might have occurred with irregular intervals and uneven amount of deposited materials. Precise calibration between exposure age and layer thickness or layer structures has not been done among lunar regolith and Itokawa regolith samples, especially for very thin layers. Hence, it seems difficult to estimate the timing of whisker formation from the features of very thin deposition layers. When whiskers are covered by deposition layers, we can only say that the deposition layers appear to have accumulated after the whisker growth (line. 99).

Q.11. Fig 2f in the caption is as a darkfield image but high Z elements id darkfield images are bright. Both the Fe and the Pt are dark.

A.11. The reviewer is maybe confused with this image. Fig 2f is a dark-field image using low angle diffracted electrons, which do not provide chemical information. This image was not obtained using high angle scattered electrons, which images are sensitive to the Z-contrast.

Q.12. Fig 3c the caption says TEM-BF image but the image label states correctly that it is a HAADF image.

A.12. The reviewer's comment is correct. We have changed the text.

Q.13. Also in this image, the label for the scale bar is either too light to see or is not there. And last, the caption for image 3d says the SAED pattern from the circled area in 3c but there is no area in 3c circled that I can see.

A.13. We have corrected the scale bar and added a circle in Fig. 3c.

Q.14. Suggestions for different word choices: iron monosulphide can just be called iron sulfide (abstract); relicted is a better word for heirloom (introduction line 50), which is used to describe grains leftover from the early solar system; (line 106) the authors use the word segregation but I think they mean coalescence;

A.14. We have changed the words "iron monosulphide" to "iron sulphide" (abstract), "heirloom" to "relics" (line 44), "segregation" to "coalescence" (line 107).

Q.15. lines 113-114 the authors use the terms vesicles and cavities as if they refer to different physical properties. If the two terms mean the same thing (which I think they do) I would recommend the authors not say vesicular regions and cavities or vesicles and cavities but just say vesicles; I would recommend not using modifiers like very and heavily.

A.15. We have changed the terms. We have used "open vesicles" instead of "cavities" throughout the manuscript (e.g., Line 84).

Q.16. It would also be good to make Supplementary Fig. 1. much bigger. I could barely see the features.

A.16. We have added an enlarged image of a troilite surface where a FIB section was lifted out (Supplementary Fig. 1b).

Q.17. Supplementary Fig. 4., I do not see where troilite is coexisting with olivine in the figure 4a, it would be great if the image were annotated with rim and host olivine, also the context for the compositional should be given (i.e., where do those rectangular composition maps overlay on the sample). Because I can't tell where the x-ray data are from I don't know whether the C map shows a high C amorphous rim? or just the carbon coat?

A.17. We have changed Supplementary Fig.4 in accordance with the reviewer's comment. We have shown the location of the EDS maps in Supplementary Fig. 4A and Supplementary Fig. 3C. Besides, we have clarified the layers of carbon and Pt coatings, vapour deposition rims, and the olivine substrate. C map shows just the carbon coat.

Reviewer #2:

Q.1. Under the "Mechanisms of whisker formation" section, the authors quickly rule out formation by condensation. I suggest that this mechanism needs a more thorough treatment. The work cited in references 28-30 provides a solid foundation for their preferred mechanism. However, whisker growth via condensation has been discussed in the meteorite literature (e.g., Bradley, John P., Ralph P. Harvey, and Harry Y. McSween Jr. "Magnetite whiskers and platelets in the ALH84001 Martian meteorite: Evidence of vapor phase growth." *Geochimica et Cosmochimica Acta* 60.24 (1996): 5149-5155.; Fries, Marc, and Andrew Steele. "Graphite whiskers in CV3 meteorites." *Science* 320.5872 (2008): 91-93.; Bradley, John P., Donald E. Brownlee, and D. R. Veblen. "Pyroxene whiskers and platelets in interplanetary dust: evidence of vapour phase growth." *Nature* 301.5900 (1983): 473.). Furthermore, micrometeorite impacts into Itokawa grains are known to deposit energy into a small volume of the asteroid surface, causing melting, vaporization, ionization, and recondensation of the target and impactor material (M. S. Thompson et al., *Microchemical and structural evidence for space weathering in soils from asteroid Itokawa. Earth, Planets and Space* 66.1, 89 (2014). Thus, it is not so

straightforward to dismiss vapor growth as a potential source mechanism. The authors should at least reference this other work and compare it to the preferred whisker growth mechanism more thoroughly.

A.1. We agree that our statement for whisker growth was not sufficient. In accordance with the reviewer's comments, we have referred prior reports about extraterrestrial whiskers and have added the text discussing the possibility of whiskers being precipitates from vapours (or liquids) in various possible conditions (lines 117-141). Because of scarce vapour-deposition-layers on grains and the lack of structures suggesting preferential iron condensation, it appears unlikely that the whiskers are condensed materials by impact.

Q.2. For the whisker composition, it would be useful to have the quantitative Ni data for all the whisker compositions.

A.2. We have added a Supplementary table. 2 detailing the TEM-EDX data obtained for all whiskers that we could analyse well (i.e., those that did not show strong signals of surrounding sulphide and Pt cover).

Q.3. Also, I request that the authors include the Ni profiles in the whiskers that showed enrichment in this element.

A.3. Unfortunately it is hardly possible to provide a representative Ni profile, because the one whisker with elevated Ni concentrations shows thickness variations (particularly at its edges) and a rather patchy distribution of Ni. To make this more obvious, we have modified Fig. 2d to better show the variation of Ni within this whisker.

Q.4. Other elements that are commonly associated with troilite in ordinary chondrites are Cr, P, Co, and Cu (e.g., Lauretta, D. S. "Sulfidation of an Iron–Nickel–Chromium–Cobalt–Phosphorus Alloy in 1% H₂S–H₂ Gas Mixtures at 400–1000° C." *Oxidation of Metals* 64.1-2 (2005): 1-22.; Rubin, Alan E. "Metallic copper in ordinary chondrites." *Meteoritics* 29.1 (1994): 93-98.). For these other elements, it would be important to know if they were analyzed for, as well as their detection limits. These elements could provide important clues to the formation mechanism.

A.4. The lowest detection limits we could achieve were ~0.1 wt% for Cr and P (the latter is estimated based on calculated k-factors, because we do not have an appropriate standard). Co and Cu are problematic due to the overlap of the Fe K β and Co K α peaks and spurious radiations emitted from the TEM column (pole pieces) and the sample grid to which the FIB sections were mounted (made of Cu). Their detection limits are at least one order of magnitude larger (probably two in case of Cu, but the only practical alternative were Mo grids which would have produced a strong overlap of Mo L lines on S K α).

Q.5. The authors discuss both solar-wind irradiation and solar heating as potential mechanisms for the production of the iron whiskers. These two mechanisms do not seem to be mutually exclusive. In particular, the sulfur evaporation rate is determined to assume a vacuum environment. However, this mechanism is likely to be enhanced by the presence of solar-wind implanted hydrogen. If it is not feasible to include the effect of hydrogen in the evaporation model, then this interaction should at least be discussed in the supplement section on this model.

Dante Lauretta

A.5. We agree that solar-wind irradiation and solar heating is mutually exclusive. We have clarified the contribution of the heating events (solar heating, micro- and larger-impact events) for sulphur loss when solar wind hydrogen is accumulated in troilite. Detailed calculation has been added in Supplemental discussion (Lines 647-710). As a result, we have added the following text;

Line 151-153: “Additional heating by micrometeorite bombardment and larger impact events may play a role for increasing the replacement of these gases by accelerating the diffusion rate of hydrogen atoms and H₂S in troilite. Furthermore, these heating events may raise the reaction rate and subsequent diffusion of iron atoms.”

Line 175: “Heating events might have contributed to promote the chemical reaction and/or atomic and molecular diffusion in troilite”

On the other hand, we have mentioned that selective sulphur loss cannot be explained solely by heating events on the asteroid Itokawa (Lines 158-173).

Reviewer #3 (Remarks to the Author):

Q1. Lines 23-24 (abstract) state "Whisker growth is a novel and unexpected aspect of space weathering." This is incorrect, whisker growth associated with space weathering has been widely reported, mostly in the context of space weathering on the lunar regolith, for the past 50+ years. Many papers are published on the topic yet none of them are cited in this manuscript. Perhaps the most relevant paper is by Carter 1973 who describes metallic iron whiskers on the moon, interpreted to result from micrometeoroid impacts.

Hibbs, A. R. A hypothesis that the surface of the moon is covered with needle crystals. Icarus, 2, 181-186 (1963).

Brownlee et al., Whiskers on the moon. In Analysis of Surveyor 3 material and photographs returned by Apollo 12. NASA SP-284, 295 pages, published by NASA, Washington, D.C., 1972, p.236

Carter, J. L. VLS (Vapor-Liquid-Solid): Newly Discovered Growth Mechanism on the Lunar Surface? Science 181, 841-842 (1973).

Steel, A. et al., Graphite in an Apollo 17 Impact Melt Breccia. Science 329, 51 (2010).

A.I. We have referred the published reports about lunar samples and have compared whiskers on Itokawa and those found on Lunar samples, and whiskers in other extrasentential materials (lines 117-141). In lines 194-196, we mentioned that the net growth mechanism of iron whiskers on Itokawa appear to be unique among the extraterrestrial whiskers, which have been widely considered as vapour condensates. Accordingly, we have corrected the abstract as follows.

Line 25: "Whisker growth by ion irradiation is a novel and unexpected aspect of space weathering"

REVIEWERS' COMMENTS:

Reviewer #2 (Remarks to the Author):

The revised manuscript largely addresses the concerns about alternate mechanisms for whisker growth. This statement is highly inaccurate "Itokawa's young formation age of $\sim 1.3-2.3$ Ga²⁸⁻³⁰ excludes the solar nebular gas as a vapour source." The papers referenced in 28-30 refer to impact heating ages. Itokawa is composed of LL5 chondrite-like material. The mineral formation ages of these meteorites are >4.56 Ga, so formation in the solar nebula cannot be ruled out with this argument. Other than this concern, the authors have done a good job reviewing other whisker growth in ET materials and providing convincing support for their preferred origin.

Reviewer #4 (Remarks to the Author):

The authors are to be commended for a thorough revision of the paper in response to the reviews. I agree with both the thrust of the initial reviews - that a broader discussion of the implications of this work for space weathering on airless bodies is needed - and the reviewers response to those. In particular, the authors make a compelling case that the iron whiskers are, in fact, related to the sulfides on which they are deposited and not vapor deposition. The authors are to be congratulated for the effort in review.

Reviewer #5 (Remarks to the Author):

In the abstract, I found that the positioning of the study is a bit confusing. The sulfur depletion on asteroids is positioned, with a consequence on S delivery on Earth. The process that is described here does not affect the overall balance of sulfur that is delivered to Earth, at least if the delivery was provided by asteroids. The modification only affects the extreme surface and do not modify the overall partitioning of S in the interplanetary space if big object are involved (like asteroid) - See also lines 64-65, and lines 231- 233.

Line 37 : it is said about the S depletion on EROS « : the depletion mechanism is however yet not understood". I do not think it is correct because an explanation is given Loeffler et al. 2008 (ref 47). Their conclusion is that the low abundance of sulfur is caused by space weathering.

Lines 159-174, about the role of solar heating. Based on the study of space weathered silicates on Itokawa, some authors claimed that it is possible that Itokawa had experienced subsolar temperatures on the order of 650 to 700 K (Harries et al. 2014, ref 15). It is much more than the 400 K mentioned here. Maybe the possibility of high T exposure for Itokawa can be discussed here because it could explain well the kinetics of the decomposition and the atomic mobility required for the whisker growth.

Comment. We are very grateful for the reviewers provided by the editors. Here is a point-by-point response to the reviewers. We have highlighted the changes within the main manuscript.

Reviewer #2 (Remarks to the Author):

Q.1. The revised manuscript largely addresses the concerns about alternate mechanisms for whisker growth. This statement is highly inaccurate "Itokawa's young formation age of ~1.3-2.3 Ga²⁸⁻³⁰ excludes the solar nebular gas as a vapour source." The papers referenced in 28-30 refer to impact heating ages. Itokawa is composed of LL5 chondrite-like material. The mineral formation ages of these meteorites are >4.56 Ga, so formation in the solar nebula cannot be ruled out with this argument. Other than this concern, the authors have done a good job reviewing other whisker growth in ET materials and providing convincing support for their preferred origin.

A.1. We correct the explanation as follows.

Line 125: "The majority of Itokawa grains display fractured surfaces that formed by the fragmentation of larger regolith grains¹², observed troilite surfaces are no exception (Fig. 1). Hence, whiskers on the troilite must have developed after the formation of the regolith grains. If iron whiskers were produced by condensation from a vapour, then Itokawa's young formation age of ~1.3-2.3 Ga³²⁻³⁴ excludes the solar nebular gas (~4.5 Ga ago) as a vapour source being in contact with Itokawa's surface regolith. The origin of iron whiskers is therefore not comparable to that of graphite whiskers reported in CV chondrites, which are thought to have condensed from the nebular gas³⁵."

Reviewer #5 (Remarks to the Author):

Q.1. In the abstract, I found that the positioning of the study is a bit confusing. The sulfur depletion on asteroids is positioned, with a consequence on S delivery on Earth. The process that is described here does not affect the overall balance of sulfur that is delivered to Earth, at least if the delivery was provided by asteroids. The modification only affects the extreme surface and does not modify the overall partitioning of S in the interplanetary space if big objects are involved (like asteroid) – See also lines 64-65, and lines 231- 233.

A.1. We added an additional explanation for the sulphur delivery to the Earth as follows.

Line 214: "Effective solar wind irradiation may also occur in small interplanetary dust due to their high surface-to-volume ratio. Sulphur loss by the irradiation could contribute to the low sulphur abundance of the Earth⁹, if irradiated small grains were involved in the accretion of the Earth⁵⁶."

Q.2. Line 37 : it is said about the S depletion on EROS « : the depletion mechanism is

however yet not understood”. I do not think it is correct because an explanation is given Loeffler et al. 2008 (ref 47). Their conclusion is that the low abundance of sulfur is caused by space weathering.

A.2. We exclude the sentence “the depletion mechanism is however yet not understood” and refer Loeffler et al. 2008 in line. 36.

Q.3. Lines 159-174, about the role of solar heating. Based on the study of space weathered silicates on Itokawa, some authors claimed that it is possible that Itokawa had experienced subsolar temperatures on the order of 650 to 700 K (Harries et al. 2014, ref 15). It is much more than the 400 K mentioned here. Maybe the possibility of high T exposure for Itokawa can be discussed here because it could explain well the kinetics of the decomposition and the atomic mobility required for the whisker growth.

A.3. The possibility of a former orbit of Itokawa closer to the Sun appears not to be too unrealistic. The temperature increase of Itokawa’s surface could have contributed to the sulphur loss rate. Hence, we add the description as below.

Line 156. “The acceleration of the reaction can also happen by solar heating when the asteroid has an orbit with a lower heliocentric distance¹⁶.”

Line 177. “Increased solar heating at possible lower heliocentric distances¹⁶ might be insufficient for a fast recession of the troilite surface (Supplementary Note 6).”

Lat part in Supplementary Note 6. “A former orbit of Itokawa closer to the Sun appears possible, but to make heating the only mechanism for whisker growth it’s surface temperature must have increased to above ~433 °C, which corresponds to an extreme orbital evolution¹⁶ currently not backed by models. As the evaporation coefficient α is probably lower than unity under realistic kinetic conditions, solar heating alone might not have caused the whisker growth.”